# Molecular Mechanisms of Drug Resistance in *Staphylococcus aureus*

**DOI:** 10.3390/ijms23158088

**Published:** 2022-07-22

**Authors:** Beata Mlynarczyk-Bonikowska, Cezary Kowalewski, Aneta Krolak-Ulinska, Wojciech Marusza

**Affiliations:** 1Department of Dermatology, Immunodermatology and Venereology, Medical University of Warsaw, Koszykowa 82a, 02-008 Warsaw, Poland; cezary.kowalewski@wum.edu.pl; 2Academy of Face Sculpting, Jana Kazimierza 11B, 01-248 Warsaw, Poland; anetau80@tlen.pl

**Keywords:** *Staphylococcus aureus*, HA/CA/LA-MRSA clones, SCC*mec*, mechanisms of drug resistance

## Abstract

This paper discusses the mechanisms of *S. aureus* drug resistance including: (1) introduction. (2) resistance to beta-lactam antibiotics, with particular emphasis on the *mec* genes found in the *Staphylococcaceae* family, the structure and occurrence of SCC*mec* cassettes, as well as differences in the presence of some virulence genes and its expression in major epidemiological types and clones of HA-MRSA, CA-MRSA, and LA-MRSA strains. Other mechanisms of resistance to beta-lactam antibiotics will also be discussed, such as mutations in the *gdpP* gene, BORSA or MODSA phenotypes, as well as resistance to ceftobiprole and ceftaroline. (3) Resistance to glycopeptides (VRSA, VISA, hVISA strains, vancomycin tolerance). (4) Resistance to oxazolidinones (mutational and enzymatic resistance to linezolid). (5) Resistance to MLS-B (macrolides, lincosamides, ketolides, and streptogramin B). (6) Aminoglycosides and spectinomicin, including resistance genes, their regulation and localization (plasmids, transposons, class I integrons, SCC*mec*), and types and spectrum of enzymes that inactivate aminoglycosides. (7). Fluoroquinolones (8) Tetracyclines, including the mechanisms of active protection of the drug target site and active efflux of the drug from the bacterial cell. (9) Mupirocin. (10) Fusidic acid. (11) Daptomycin. (12) Resistance to other antibiotics and chemioterapeutics (e.g., streptogramins A, quinupristin/dalfopristin, chloramphenicol, rifampicin, fosfomycin, trimethoprim) (13) Molecular epidemiology of MRSA.

## 1. Introduction

The species of staphylococcus that is commonly associated with increasing bacterial resistance to antibiotics is *Staphylococcus aureus*. It is currently included in the ESKAPE group (*Enterococcus faecium*, *Staphylococcus aureus*, *Klebsiella pneumoniae*, *Acinetobacter baumannii*, *Pseudomonas aeruginosa*, and *Enterobacter* species), a group of the most important bacteria involved in infections and characterized by multidrug resistance [1].

*S. aureus* occupies a special place among the above-mentioned species due to its relatively high virulence on the one hand, and great plasticity on the other hand, enabling it to adapt to various environmental conditions. *S. aureus* strains have evolved resistance mechanisms to almost all antimicrobial drugs used in treatment. The most important is resistance to drugs most commonly used in the treatment of Gram-positive infections, i.e., beta-lactams, glycopeptides, and oxazolidinones.

Most problems were caused by MRSA strains, which led to infections that were difficult to treat [2]. The first MRSA strains appeared in 1960–1961 and were characterized by resistance to all the beta-lactam antibiotics then used in the treatment. It was not until 2010 that two cephalosporins, ceftobiprole and ceftaroline, active against MRSA and MRS-CN strains, were introduced. Within a short time, strains resistant to these two drugs also emerged. MRSA strains spread in the 1970s and 1980s when cephalosporins were used on a massive scale in hospitals. For many years, MRSA strains were equated with hospital-acquired MRSA (HA-MRSA, later expanded to health care-associated MRSA). Some authors use the term HCA-MRSA for this group of strains. In the 1990s, new MRSA strains started to appear, associated with infections in non-hospitalized patients (Community acquired CA-MRSA) [2] and at the beginning of the 21 st century LA-MRSA (livestock-associated MRSA) was described. Resistance to beta-lactams occurring in MRSA strains and many MRS-CN, is associated with the presence in the bacterial genome of transferable genomic islands (GI, genomic islands), called SCC*mec* (staphylococcal chromosomal cassette *mec*), where the *mec* gene determines resistance to methicillin. These islands evolve rapidly and contain many mobile genomic elements. Within the different types of SCC*mec*, there may be *mecA* or *mecC* genes and resistance genes to other groups of antibiotics such as aminoglycosides, macrolides, lincosamides, streptogramins B and tetracyclines (MLS-B) [3].

In the 90′s, strains intermediate-susceptible to vancomycin and other glycopeptides were also described (VISA—vancomycin intermediate *S. aureus*, GISA—glycopeptide intermediate *S. aureus*). According to current European EUCAST criteria, strains previously classified as VISA are now classified as VRSA (MIC vancomycin > 2 mg/L). In 2002, the first vancomycin-resistant *S. aureus* (VRSA) strains were detected, possessing the vanA operon in the Tn1546 transposon. Tn1546, which determines resistance to vancomycin, was described in *Enterococcus* genus since 1988. Subsequent studies have shown that Tn1546 is not expressed in most strains of staphylococci.

Resistance to glycopeptides encoded by the VanA operon (usually vancomycin MIC ≥ 16 mg/L) was expressed more frequently in *S. aureus* strains with mutation of the modification-restriction system and/or possessing the pSK41-like conjugation plasmid (factors that increase the frequency of VanA operon conjugation). Therefore, only about a dozen VRSA strains with the VanA operon have been described on the working scale [4,5,6,7].

In 2006, the first class I integrons, previously described in Gram-negative bacteria, were detected in staphylococci, initially found in coagulase-negative staphylococci and later in *S. aureus*. The cassette genes functioning in these integrons that determine resistance to streptomycin (*aadA2*, *aadA5*), chloramphenicol (*cmlA1*) and trimethoprim (*dfrA12*, *dfr17*) have also been described [8,9]. These are the same gene cassettes as found in Gram-negative bacilli. The spread of other gene cassettes and integrons in staphylococci and other Gram-positive bacteria may pose risks associated with the potential for rapid, interspecies, and intergeneric exchange of resistance genes, as well as the acquisition by Gram-positive bacteria of genes found in Gram-negative bacteria.

The efficacy of antimicrobial drugs in the treatment of *S. aureus* infections is not only related to the resistance or lack of resistance to a given drug, but may depend on many other factors such as the growth phase of the bacterium (logarithmic growth phase or stationary phase), the localization of the infection (drugs reach different concentrations in different compartments of the body), biofilm formation, and many others. The treatment strategy should also be correlated with the type of infection (epidemic, nosocomial, chronic, etc.), which in turn may be related to the presence or absence of a specific pathogenicity factor.

*S. aureus* has an enormous potential for pathogenicity. The virulence factors occur at different frequencies in different clones of bacteria. Some pathogenicity factors have several functions and can be included in several groups simultaneously. Frequently virulence genes occur together with drug resistance genes on the same genetic elements. However, the mere presence of a pathogenicity factor gene does not always equate to its expression, which is subject to one or more regulatory systems present in the bacterial cell, such as the Agr/Sar system [10] These systems can also affect the expression and stability of genes that determine antibiotic resistance.

## 2. Resistance to Beta-Lactam Antibiotics

Beta-lactam antibiotics target certain enzymes (transpeptidases, transglycosylases, carboxypeptidases) involved in the synthesis of peptidoglycan, a key element of the bacterial cell wall. Proteins inactivated by beta-lactams are referred to as PBPs (penicillin binding proteins). Inactivation of specific PBPs leads to bacterial cell death. Several mechanisms of resistance to beta-lactam antibiotics are known in *S. aureus*. These are: synthesis of a new, additional PBP called PBP2A (PBP2a, PBP2′), synthesis of beta-lactamases, and mutations in PBP genes. The mechanisms are presented in Figure 1.

### 2.1. Synthesis of Beta-Lactamases

In staphylococci, the mechanism involving the synthesis of beta-lactamases (penicillinases) is very common. These are enzymes with a narrow substrate spectrum, including the so-called beta-lactamase-sensitive penicillins, including natural penicillins and aminopenicillins. Beta-lactamases synthesized by *S. aureus* are classified as group 2a by Bush [11] and class A by Ambler. They are synthesized by most MRSA but also very frequently by MSSA. These enzymes are encoded by the *blaZ* gene usually within the *blaI-blaR1-blaZ* operon found in numerous plasmids and transposons (e.g., pI258, pII147, Tn552, Tn4002, Tn4201, SCC*mec* type XI) [12,13]. Four variants of staphylococcal β-lactamase (A-D) can be distinguished by serotype and currently activity profile [14]. Single strains of the so-called BORSA (borderline oxacillin-resistant *S. aureus*) have been described, in which the synthesized beta-lactamase had an extended spectrum and conditioned an oxacillin MIC of 4–8 mg/L [15].

### 2.2. PBP2A Synthesis (Methicillin-Resistance)

In *S. aureus*, resistance associated with the synthesis of the novel PBP2A protein is of greatest importance. Genes encoding additional PBPs are located within the SCC*mec* chromosomal cassettes and are transmitted by conjugation or transduction. The criterion of methicillin resistance is MIC oxacillin > 2 mg/L in *S. aureus*, *S. lugdunensis* and *S. saprophyticus* and MIC oxacillin > 0.25 mg/L for the remaining species of the genus *Staphylococcus* according to EUCAST [16] and according to CLSI for *S. aureus* and *S. lugdunensis* MIC oxacillin ≥ 4 mg/L and for the remaining species of the genus *Staphylococcus* MIC oxacillin ≥ 1 [17]. The cefoxitin resistance criterion is also used: MIC > 4 mg/L for *S. aureus* and *S. lugdunensis* and MIC > 8 mg/L for *S. saprophyticus* according to EUCAST [16] and MIC ≥ 8 mg/L for *S. aureus* according to CLSI [17].

Strains possessing PBP2A show resistance to all therapeutic beta-lactam antibiotics except ceftobiprole and ceftaroline and are referred to as methicillin-resistant *Staphylococcus aureus*, MRSA. The mechanism of methicillin resistance is closely related to PBP proteins involved in peptidoglycan synthesis. *S. aureus* methicillin-susceptible (MSSA) strains synthesize: PBP1 (744 amino acids, two domains: transpeptidase and dimerization), PBP2 (727 amino acids, three domains: transpeptidase, transglycosylase and carboxypeptidase), a key protein of *S. aureus* and inactivation of the transpeptidase function of this protein leads to cell death, PBP3 (691 amino acids, two transpeptidase and dimerization domains) and PBP4 (491 amino acids, D-alanyl-D-alanine carboxypeptidase domain) [18]. MRSA strains additionally synthesize PBP2A (PBP2a or PBP2′) of 668 amino acids, which replaces the transpeptidase function of PBP2 and takes over the transpeptidase function of other, inactivated PBPs. Beta-lactam antibiotics inactivate the PBP2 transpeptidase domain, while the PBP2 transglycosylase domain remains active and, in the case of MRSA strains, cooperates with the PBP2A transpeptidase [19]. PBP2A, together with about 40 other enzymes, is involved in the formation of pentaglycan bridges, between L-lysine (position 3 of the pentapeptide) of one chain and D-alanine (position 4 of the pentapeptide) of the other peptidoglycan chain. This enables the synthesis of the cell wall. PBP2A expression can be inducible or constitutive. PBP2A synthesis is usually associated with the presence *mecI-mecR1-mecA* or *mecI_c_-mecR1_c_-mecC* operon (inducible expression) or Δ*mecR1-mecA* (constitutive expression), located in staphylococcal chromosomal cassette mec (SCC*mec*) cassettes, found in the chromosome and classified as so-called genomic islands (GI) [20]. The *mec* genes are transferred to susceptible strains together with the entire SCC*mec* structure.

In bacteria belonging to the *Staphylococcaceae* family, 8 genes encoding the PBP2a transpeptidase have been described. The *mecA* gene has been detected in *S. aureus*, *S. pseudintermedius*, *Mammallicoccus fleuretti* (former name *Staphylococcus fleuretti*), *Mammallicoccus vitulinus* (former name *Staphylococcus vitulinus*), *Mammallicoccus sciuri* (former name *Staphylococcus sciuri*) and in many other species of the genus *Staphylococcus* currently comprising 64 species [21]. The *mecA1* gene has been described in *M. sciuri*. It is considered a precursor of the *mecA* gene in *S. aureus*, and the *mecA2* gene. *M. sciuri* oxacillin-susceptible (OS-MRSS, MIC oxacillin = 1 mg/L; according to current criteria, these are methicillin-resistant strains) and MRSS strains were described (they had two genes: *mecA1* and *mecA*) [22]. The *mecA2* gene was described in *M. vitulinus* and *S. capitis*, the *mecB* gene in *Macrococcus caseolyticus* and *S. aureus*, the *mecC* gene in *S. aureus* and *M. sciuri*, the *mecC1* gene in *S. xylosus*, the *mecC2* gene in *S. saprophyticus*, and the *mecD* gene in *M. caseolyticus* [23,24]. Evolutionarily, the *mecA* gene present in *S. aureus* is likely derived from the coagulase-negative staphylococcal group *S. sciuri* which includes *S. sciuri*, *S. fleurettii*, *S. vitulinus*, among others [24]. In 2018, the expression of *mecB* and *mecD* genes previously described in *Macrococcus caseolyticus* isolated from dogs and cattle were described in *S. aureus* [25]. The *mecB* gene was described to be present in plasmid pSAWWU4229 1 of 84599 bp. This plasmid also encoded resistance genes to aminoglycosides, macrolides and tetracyclines. [25].

#### SCCmec Chromosomal Cassettes

Fourteen major types of SCC*mec* have been described and sequenced, and within the major types are several subtypes [26,27,28,29,30,31,32]. The described SCC*mec’s* range in size from 21 to 82 thousand nucleotides. In the structure of SCC*mec* cassettes, five regions are usually specified. The *orfX* included in the schematic is a site in the chromosome into which SCC*mec* is incorporated by unauthorized recombination and is not part of SCC*mec* (Figure 2).

The division of *SCCmec* into main types is related to the type of *ccr* chromosomal recombinase gene complex: *ccrA*, *ccrB*, *ccrC*. Within *ccrA* we distinguish *ccrA1*, *ccrA2*, *ccrA3* and *ccrA4* which are found in *S. aureus*. *ccrA1* has also been described in *S. hominis* and *S. saprophyticus*, *ccrA4* has been described in *S. epidermidis*, in addition *ccrA5* has been described in *S. pseudintermedium* and *ccrA7* has been described in *S. sciuri*. Within *ccrB* there are *ccrB1*, *ccrB2*, *ccrB3*, *ccrB4*, and *ccrB6* which are found in *S. aureus*. *ccrB1* is described in *S. hominis*, *ccrB3* is described in *S. pseudintermedius*, *ccrB4* is described in *S. epidermidis*, and *ccrB6* and *ccrB7* are described in *S. saprophyticus*. Among *ccrC1*, allotypes have been described, with 1, 2, 3, 4, and 8 found in *S. aureus*, 5 and 6 described in *S. haemolyticus*, allotype 7 described in *S. epidermidis*, and allotype 9 described in *S. saprophyticus* [33,34,35].

The second important criterion for the division of SCC*mec* is the class of the mec region. The following classes are distinguished: A, B, B2, C1, C2, D, and E. The different classes differ in the degree of deletion of *mecI-mecR (*Δ*mecR1*), regulatory genes and their proximity and distance from the complete or reduced (Δ,Ψ) insertion sequences IS431, IS1182 and IS1272 [32,33,34,36]. The division of SCC*mec* into subtypes is based on the mec region subclasses and the structure of the J1, J2, and J3 regions [32,37,38].

Complexes of gene *mec* and *ccr* [33,34,38] are shown in Table 1 and Table 2.

In *S. pseudintermedius*, *ccrA5* and *ccrB3* (*ccr* gene complex 6) were described in SCC*mec* KM241 and *ccrC6* in SCCmec NA45 [41]. *ccrA1* and *ccrB4* were described in *S. saprophyticus* and *ccrA5/ccrB3* in *S. hominis*, *S. haemolyticus* and *S. cohnii*; *ccrA1/ccrB1* and *ccrC1* in *S. cohnii*, *ccrA2/ccrB2* and *ccrC1* and *ccrA4/ccrB4* and *ccrC1* in *S. epidermidis* and *ccrA7/ccrB3* in *M. sciuri* [42].

The types of SCC*mec* chromosomal cassettes found in different MRSA clones according to [26,43,44,45,46,47,48,49,50,51,52] are shown in Table 3.

Multiple SCC*mec* subtypes have been described (strain-GenBank accession No.): IIa (N315, Mu50, MRSA252, JH1-NC_002745, NC_002758, BX571856, NC_009632); IIb (JCSC3063-AB127982); IIc (AR13.1.3330. 2-AJ810120); IId (RN7170-AB261975), IIe (JCSC6833-AB435013) [32,37,43,53,54,55,56]; IIIA (HU25-AF422651, AF422696); IIIB (HDG2) [30,37]; IVa (CA05-AB063172); IVb (8/6-3P-AB063173); IVc (MR108, MRSA NN424-AB096217, KX211998. 1); IVd (JCSC4469-AB097677); IVE (AR43/3330. 1-AJ810121.1); IVF; IVg (M03-68-DQ106887); IVh (HO50960412/EMRSA15-HE681097); IVi (JCSC6668/CCUG41764 -AB425823); IVj (JCSC6670/CCUG27050-AB425824); IVk (45394F-GU122149); IVl (NN50, SI1-AB633329. 1, LC425379.1); IVm (JCSC8843-AB872254); IVn (ST93-KX385846. 1); IVo [28,32,37,57,58,59,60,61,62,63,64,65,66]; Va (WIS(WBG8318)-AB121219); Vb (TSGH17, JCSC7190), PM1, JCSC5952-AB512767, AB462393, AB478780); Vc (S0385, JCSC6944-AM990992, AB505629) [26,45,52,67].

### 2.3. Mutation-Dependent Modification of PBP Proteins

Mutations in genes encoding PBP2 and PBP4 causing oxacillin resistance are very rare and strains are described as MODSA (modified penicillin-binding protein *S. aureus*) or MRLM (methicillin-resistant lacking *mec*). The most frequent causes are mutations in the promoter region of the *pbp4* gene and in the *gdpP* (phosphodiesterase c-di-AMP regulator) and *yjbH* (disulfide stress effector) genes, conditioning the overproduction of PBP4 protein [68,69,70]. Resistance to ceftobiprole was described in *S. aureus* strain CRB (a derivative of strain COL). The strain lacked the *mecA* gene, but had substitution in PBP4 protein (E183A, F241R), GdpP signaling protein (N182K) and the ArcB/D/F cations efflux pump protein (I960V). The presence of the above-mentioned mutations resulted in CRB strain resistance to beta-lactam antibiotics: ceftobiprole (MIC = 128 mg/L), ceftriaxone (MIC = 256 mg/L), cefazolin (MIC > 256 mg/L), nafcillin (MIC = 128 mg/L), ampicillin (MIC = 256 mg/L). Only the MIC of cefoxitin was relatively low (8 mg/L) [71]. Substitutions in PBP4 protein (N138K or I, R200L, T201A, F241L, and H270L) determining resistance to ceftobiprole and ceftaroline in MSSA strains were also described [72]. In MRSA strains, resistance to ceftobiprole and ceftaroline is caused by mutations in the *mecA* gene. Strains harboring a D239L, S225R, N146K substitution or a 259–260 insertion in PBP2A had ceftobiprole MICs of 4–8 mg/L [73,74]. Amino acid substitutions in the PBP2A protein such as L357I, E447K, I563T, and S649A in the BND (penicillin-binding domain) and substitutions N104K, V117I, N146K and A228V located outside the BND have also been shown to be responsible for ceftaroline resistance in MRSA strains [75,76]. Breakpoint of resistance to ceftaroline is MIC > 1 mg/L (pneumonia) and MIC > 2 mg/L (other than pneumonia) according to EUCAST [14] and MIC ≥ 8 mg/L according to CLSI [17]. Breakpoint of ceftobibrole resistance is reported only by EUCAST (MIC > 2 mg/L) [16].

## 3. Resistance to Glycopeptides and Lipoglycopeptides

Glycopeptides and lipoglycopeptides, like beta-lactams, are bactericidal and their mechanism of action is through inhibition of peptidoglycan synthesis, but their target and exact mechanism of action is quite different. Glycopeptides and lipoglycopeptides form bonds with the dipeptide D-Ala-D-Ala within GlcNAc-β-(1,4)-MurNAc-pentapetide, the precursor of peptidoglycan. Oritavancin also shows affinity for binding to the D-ala-lactate dimer within GlcNAc-β-(1,4)-MurNAc-pentapetide found in strains expressing the VanA operon. These processes occur outside the cytoplasmic membrane. Moreover, lipoglycopeptides bind to the bacterial cytoplasmic membrane and cause rapid, concentration-dependent depolarization of the cytoplasmic membrane, increased permeability and leakage of cellular ATP and K+ ions leading to cell death. Moreover, lipoglycopeptides probably inhibit transglycosidases which are involved in the polymerization of uncrosslinked peptidoglycan precursors [77].

The criteria for *S. aureus* resistance to glycopeptides used in Europe (EUCAST) and in the USA (CLSI) differ significantly. According to EUCAST, *S. aureus* resistant to vancomycin (VRSA) or teicoplanin (TRSA) have MIC > 2 mg/L [16] and according to CLSI breakpoint for VRSA is MIC ≥ 16 mg/L and for TRSA is MIC ≥ 32 mg/L [17]. Breakpoints of resistance to lipoglycopeptides such as dalbavancin, oritavancin and telavancin are MIC > 0.125 mg/L according to EUCAST [16]. CLSI reports only breakpoint sensitivity, MIC ≤ 0.12 mg/L for oritavancin and telavancin and MIC ≤ 0.25 mg/L for dalbavancin [17].

Resistance to glycopeptides has been best described in enterococci. It can be conditioned by different operons named after the ligase they encode: VanA, VanB, VanC, VanD, VanE, VanG, VanL, VanM and VanN. In *S. aureus*, resistance to high concentrations of glycopeptides occurs very rarely and is then conditioned by the VanA operon derived from *Enterococcus* spp (Figure 1). It consists, as in *Enterococcus* spp., in the synthesis of an altered precursor of the cell wall, which instead of the terminal group D-Ala-D-Ala has D-Ala-D-lactate [78].

The VanA operon that determines resistance to vancomycin (MIC 64–1024 mg/L) and teicoplanin (MIC 16–512 mg/L) is composed of 7 genes, (*van*R_A_S_A_H_A_AX_A_Y_A_Z_A_) (Figure 3) located at Tn1546 and was described in *E. faecalis*, *E. faecium*, *E. gallinarum*, *E. casseliflavus*, *E. avium*, *E. durans*, *E. mundtii* and *E. rafinosus* [76]). The Van A operon shows inducible expression mediated by two regulatory genes *vanR_A_* (regulator) and *vanS_A_* (sensor, a signal histidine kinase located in the cytoplasmic membrane). vVanS_A_ sensor activation is caused by both vancomycin and teicoplanin. Alterations in the central *van*RSHA region of the Tn1546 transposon can result in varying teicoplanin MICs from >256 to <4 mg/L [79].

VRSA conditioned by the VanA operon emerged in the US in 2002 and resulted from the transfer of *Enterococcus* spp. Tn1546 was incorporated into a plasmid and into MRSA strains. Single VRSA or VRSA/TRSA strains have been described in the USA and Asia [4,5]. The first VRSA, Michigan (MI-VRSA) had high resistance to vancomycin and teicoplanin and a 57.9 kb plasmid, similar to pSK41, with a Tn1546-like insertion [5], while the one originating from Pennsylvania (PE-VRSA) showed resistance only to vancomycin and easily lost the plasmid with the VanA operon [80]. MRSA strains with a VanA operon that was not expressed were also detected, which are difficult to detect by routine diagnostics [81]. Most VRSA strains have been found to belong to the common MLST (multilocus sequence typing) CC5 clonal complex [6,7]. In addition, some VRSA with the VanA operon had a mutation in the *hsdR* gene encoding Sau1 (1 modification-restriction system) [5]. In several *S. aureus* strains carrying the *vanA* gene (irrespective of its expression), the presence of other resistance genes probably derived from *E. faecium*: *ermB* (MLS-B resistance), *aadE*(ant(6)-Ia) (streptomycin resistance), *sat4* (streptothricin resistance), *aphA-3* (Aph(3′)-IIIa, aminoglycosides resistance), *msrA* (efflux macrolides and streptogramins B), *aac(6′)-aph(2″)-Ia* (resistance to aminoglycosides), and *tet(S)* and *tet(U)*, (resistance to tetracyclines) [82]. The 57.9 kb conjugative plasmid pLW043 (GenBank AE017171) described in *S. aureus* contained, among other things, the VanA operon in the Tn1546 transposon, the beta-lactamase operon, the *aac(6′)/aph(2″)* gene for bifunctional aminoglycoside transferase, and the *dfrA* gene for dihydrofolate reductase that determines resistance to trimethoprim [82].

In addition to VRSA (GISA) strains, the 2022 CLSI criteria specify VISA (MIC vancomycin 4–8 mg/L) and hVISA (heterogeneous VISA) strains. The vancomycin MIC for hVISA strains is usually 1.5–3 mg/L and hVISA qualification requires confirmation by population PAP-AUC analysis. The PAP-AUC ratio was interpreted as follows, <0.9 as vancomycin-susceptible *S. aureus* (VSSA), ≥0.9 as hVISA phenotype, >1.3 as vancomycin-intermediate *S. aureus* (VISA) [83].

The reduced sensitivity to vancomycin is probably due to different genome rearrangements. According to the 2022 EUCAST criteria [16] currently in force in Europe, we designate *S. aureus* strains as vancomycin resistant (MIC > 2 mg/L) or sensitive (MIC2 mg/L). Sensitivity to vancomycin according to EUCAST is equivalent to the sensitivity to lipoglycopeptides: dalbavancin, oritavancin and telavancin (MIC ≤ 0.125 mg/L).

VISA strains (VRSA according to EUCAST criteria) were mostly described within HA-MRSA clones such as ST5-II/III, ST8-II, ST239-III, ST241-III, ST247-IA. VISA was also described in ST1-IV, ST30-IV, ST59-IV, ST72-IV, ST81-IV, ST45, ST228-I, ST398, ST900-III, ST1301-II [84,85,86,87]. Substitutions in VraS (S329L), MsrR (E146K), GraR (N197S), RpoB (H481Y or N), Fdh2 (A297V) proteins have been shown to be closely related to vancomycin resistance of VISA strains [88]. The accumulation of mutations in genes encoding binary regulatory systems such as WalKR (sensor protein kinase/regulator), GraSR (glycopeptide resistance-associated sensor/regulator), and VraSR (vancomycin resistance associated sensor/regulator) plays a major role in the formation of hVISA/VISA strains [75].

*S. aureus* TRSA strains sensitive to vancomycin (MIC 1.5–3 mg/L) and resistant to teicoplanin (MIC 4–32 mg/L) were also described. *S. aureus* strains are defined as resistant according to EUCAST criteria (MIC > 2 mg/L) [16] and according to CLSI criteria (MIC ≥ 32 mg/L) [17]. These were methicillin-resistant strains belonging to ST772-V, ST672-IVa and ST22-IVc. The resistance was probably caused by mutations in *tcaA* (sibstitutions in TcaA D230E or F290S), *tcaB* (substitution in TcaB Y6R), *lytS* (LytS substitutions: P315R, A318Q, A319L, I320S, andV321M), and *rhoR* (substitutions in RhoR: V186I, L144I and V535M) genes [84].

An *S. aureus* strain V036-V64 (ST5) with a MIC of 64 mg/L vancomycin obtained from a susceptible strain by multiple passages on vancomycin medium was also described. Single amino acid substitutions in 8 proteins were shown relative to the starting strain:

RimM (G16D), SsaA2 (G128A), RpsK (P60R), RpoB (R917C), WalK (T492R), D-alanyl-D-alanine carboxypeptidase (L307I), VraT (A152V), and chromosome segregation ATPase (T440I). Strain V036-V64 showed an increase in the MIC of vancomycin from 0.5 to 64 mg/L, teicoplanin from 0.5 to 3 mg/L, daptomycin from 0.25 to 4 mg/L, and telavancin from 0.047 to 0.25 mg/L (telavancin resistance according to EUCAST) [89].

In *S. aureus* there is also a phenomenon of tolerance to glycopeptides (MBC/MIC32). Such strains show sensitivity to glycopeptides but are not killed by them. Picazo et al. [90] studied the phenomenon of tolerance to vancomycin and teicoplanin in 187 MRSA strains from 41 Spanish hospitals and showed tolerance to vancomycin in 9.6% of MRSA strains and tolerance to teicoplanin in 21.9% of MRSA strains [90]. The reason for vancomycin tolerance may be the ability to form a biofilm and, in non-biofilm-forming strains, altered autolysis activity, by lysogenic conversion, for example [91,92].

## 4. Resistance to Oxazolidinones

Linezolid and tedizolid belong to oxazolidinones and act on the 23S rRNA molecule in the 50S subunit of the ribosome (inhibition of protein synthesis) and show high activity against *S. aureus* (including MSSA, MRSA, VRSA and VISA). Linezolid binds to the conserved nucleotide A2602, which is part of the V domain of the 23S rRNA in the 50S subunit, and to two proteins on the same subunit: the ribosomal protein L27, whose N-end is closely adjacent to the active center of peptidyl transferase, and the LepA [93].

*S. aureus* strains for which the value of linezolid MIC is ≥8 mg/L or tedizolid MIC is ≥2 mg/L according to CLSI [17] and linezolid MIC is >4 mg/L or tedizolid MIC is >0.5 mg/L according to EUCAST [16] are classified as resistant. Resistance to oxazolidinones may result from mutations in the *rrn5* gene, mutations in *rplC*, *rplD*, *rplV* genes encoding ribosomal proteins L3 (G152D substitution), L4 (K68Q substitution), L22, mutations and expression of *cfr* genes. These mechanisms also determine resistance to lincosamides, phenicols, streptograminA and pleuromutilin [75]. A mutation in the *rpoB* gene (A1345G; substitution in the RpoB protein D449N) was also described, which determines the resistance of *S. aureus* strain to linezolid (MIC = 8 mg/L) and tedizolid (MIC = 4 mg/L), chloramphenicol (MIC-128 mg/L), medium sensitivity to quinupristin/dalphopristin (MIC-2 mg/L) [94]. Mutation in the *rrn5* gene encoding 23S rRNA in the 50S subunit of the ribosome results in modification of the target site for linezolid within the V domain of the 23S rRNA and prevents linezolid action. Among strains isolated from clinical cases, the G2576U mutation (G2576T in rDNA) appears to be the most significant. This mutation has been found in single *S. aureus* strains [95,96]. In one linezolid-resistant *S. aureus* strain another mutation of the *rrl5* gene, namely T2500A, was found, in the absence of G2576U, G2447T mutations were also described [75,96]. It was found that for four resistant clinical isolates of *S. aureus*, characterized by linezolid MIC values of 16 mg/L (three strains) and 8 mg/L (one strain), the G2576T mutation was always present in two alleles out of five, with copy 5 (*rrn5*) always present [96].

A different mechanism of resistance to oxazolidinones involves the synthesis of an adenylyl-N-methyltransferase Cfr that causes dimethylation of adenine (A2503) within the V domain of 23S rRNA of bacterial ribosome (Figure 1). The *cfr* gene was found in plasmid p004-737X (*istAS-istBS-cfr-tnp*) of 55 kb [97]. The occurrence of Cfr has been described in *S. aureus* in the Tn556 transposon embedded in plasmid pSCFS6 (GenBank AM408573) [98,99]. Expression of the *cfr* gene causes resistance to oxazolidinones (linezolid), lincosamides, phenicoles (chloramphenicol), streptogramin A (dalfopristin) and retapamulins (pleuromutilin) [97].

Another mechanism is conditioned in *S. aureus* by ARE ABC-F (antibiotic resistance (ARE) proteins belong to the F lineage of the ABC superfamily) proteins such as OptrA conditioning resistance to linezolid and phenicols and PoxtA protein conditioning resistance to linezolid, phenicols and tetracyclines through a ribosomal protection mechanism [100,101].

## 5. Resistance to Macrolides, Lincosamides, Ketolides and Streptogramins B

Most of the macrolides, lincosamides and streptogramins B (MLS-B) show the same mechanism of action. The target site for them is a four-nucleotide rRNA fragment (in the peptidyltransferase region) within the V domain of the 23S rRNA, in the 50S subunit of the ribosome. A different mechanism is demonstrated by the macrolide antibiotic approved by the FDA for the treatment of *Clostridioides difficile* infections, fidaxomicin, which inhibits DNA-dependent RNA polymerase [102]. Resistant *S. aureus* have MIC azithromycin, clarithromycin, erythromycin, dirythromycin ≥ 8 mg/L and MIC clindamycin ≥ 4 mg/L according to CLSI [17] and MIC azithromycin, clarithromycin, erythromycin > 2 mg/L and MIC clindamycin > 0.25 mg/L according to EUCAST [16].

There are various mechanisms of *S. aureus* resistance to MLS-B antibiotics. The most common mechanism involves modification of the target site for the antibiotic. The modification is carried out by the enzymes adenylyl-N-methyltransferase Erm (erythromycin ribosome methylation) (Figure 1), dimethylating adenine 2058, which leads to resistance to all MLS-B. The gene encoding Erm methylase synthetase may be expressed in a constitutive manner, in which case strains show resistance to all MLS-B, or in an inducible manner, in which case resistance occurs only to antibiotics that are inducers of methylase synthesis, i.e., macrolides with a 14-member ring (M14) except ketolides (e.g., erythromycin, clarithromycin, oleandomycin) and 15-member ring (M15, e.g., azithromycin) [103]. Resistance to the other MLS-B requires the presence of an inducer, which may be erythromycin or another macrolide M14-15 [104]. CLSI’s introduction of an inducible MLS-B resistance test for staphylococci and streptococci into routine testing was intended to preclude the use of clindamycin in patients infected with bacteria that, although showing sensitivity to this antibiotic, can very easily become resistant to it during treatment. Inducible resistance to MLS-B in *S. aureus* is most often determined by the *ermA* or *ermC* genes [103]. The frequency of formation of constitutive variants from inducible *ermA* genes has been determined to be approximately 10^−6^–10^−8^, and for inducible *ermC* genes, the frequency is usually much higher. Causes of constitutive variants are deletions, duplications, insertions, and, relatively rarely, point mutations located in a region about 200 bp upstream of the 5′ end of the *erm* gene. Differences in frequency of constitutive variants formation are probably connected with different localization and a different number of copies per cell of *ermA* and *ermC* genes. The *ermA* gene, found in *S. aureus* in the chromosome as part of the Tn554 transposon (containing the *ermA* gene and the *spc* gene for resistance to spectinomycin), has one site of high preference and another site of 1000× less preference for integration into the chromosome. Tn554 is mostly found in one copy per chromosome. In MRSA strains, Tn554 is additionally present in SCC*mec* type II, III or VIII cassettes. The *ermA* gene has also been described in the Tn6072 transposon (GenBank GU235985) [105]. The *ermC* gene encoding a 23S rRNA (adenino 2085-N6)-dimethyltransferase (EC 2.1.1. 184) is mostly located within small plasmids such as pE194 (GenBank V01278) [106], pT48 (GenBank M19652) [107], pE5 (GenBank M17990) [108], pJR5 (GenBank L04687) [109], pA22 (GenBank X54338) [110], pJ3356::POX7 (GenBank U36911) [111], pWBG738 with a size of 2.5–5.0 kb, found in high copy number and within large conjugation plasmids, e.g., pUSA03 (37 kb), where it occurs together with the *ileS2* gene that conditions mupirocin resistance [112,113].

Regulation of *ermA* and *ermC* gene expression occurs at the translational stage. The *ermA* gene is preceded in the polycistronic strand by two genes encoding leader peptides: *pepL* and *pep1*, whereas in the case of *ermC* by one, *pep*. The bundled structures formed by rRNA *pep* prevent ribosome access to the RBS (ribosome binding site) for the *erm* gene. Therefore, in the absence of an inducer, Erm methylase synthesis does not occur. On the other hand, if an inducer molecule (M14-15.) binds to the ribosome beforehand, translation of the leader peptide is interrupted and dissociation of the ribosome does not occur. This results in a permanent bifurcation of the mRNA spliced structure, allowing ribosome access to the RBS for the *erm* gene and its translation (Figure 4) [113]. Point mutations (G98A, A137C, C140T and G205A) in the regulatory region of the *ermA* gene have been described. The *ermA* gene in which the G98A, A137C and C140T mutations were present (phenotypes 1 and 2) did not show expression of azithromycin and clindamycin resistance [114]. Other mechanisms of resistance to MLS-B have also been described in *S. aureus* strains.

Synthesis of 23S rRNA methylases other than Erm(A) and Erm(C) that modify the target site for the antibiotic: Erm(GM), (another name for Erm(Y)), the *ermGM* gene occurs in plasmid pMS97, (GenBank AB014481) [115]; Erm(B), the *ermB* gene occurs in Tn551 often described in hMRSA and VRSA chromosome and in pI258 (GenBank AB300568) [82,116,117]; Erm(F) [118]; Erm(T) [119]; Cfr conditions resistance to lincosamides and some antibiotics of other groups [82,97].

Synthesis of efflux pumps proteins such as ABC proteins Msr(A), Msr(A)/Msr(B) (M14SB resistance) [120,121,122] and MsrSA (GenBank AB013298, M14-15SB resistance) [123]; Vga(A) proteins (clindamycin and lincomycin resistance) [124].

Synthesis of MLS-B-inactivating enzymes such as macrolide phosphotransferases Mph(BM) (GenBank AB013298) and Mph(C) described in plasmids pMS97, pSR1 (GenBank AF167161) and in animal biotypes of *S. aureus* [113,125,126]; Lnu(A) nucleotidyltransferase (other names LinA, LinA1, or LinA’; lincomycin resistance) [127,128]; Ere(A), Ere(B) esterases (M14,16 resistance) [129]; and VgbA, VgbB, inactivating streptogramin B lyases [130,131,132].

Resistance to MLS-B can also result from mutations in chromosomal genes encoding ribosomal proteins, such as a mutation in the *rplV* gene encoding the L22 protein in the 50S subunit of the ribosome, which determines resistance to erythromycin, telithromycin, quinupristin, and dalfopristin in *S. aureus* [133] and a mutation in the *rplD* gene encoding the L4 protein in the 50S subunit of the ribosome that conditions resistance to erythromycin and spiramycin in *S. aureus* [134].

## 6. Resistance to Aminoglycosides and Spectinomycin

Aminoglycosides are bactericidal antibiotics that inhibit protein synthesis by interfering with the 30S subunit of the ribosome. Breakpoints of resistant *S. aureus* to gentamicin and tobramycin are MIC > 2 mg/L and to amikacin are MIC > 16 mg/L according to EUSAST [16] and gentamicin MIC ≥ 16 mg/L according to CLSI [17]. In routine diagnostics in *S. aureus*, only gentamicin resistance is determined. Gentamicin-resistant *S. aureus* strains (those having the *aacA-aphD* gene) are usually resistant to all aminoglycoside antibiotics currently used in human therapy, distreptamine derivatives.

Resistance to aminoglycosids in *S. aureus* may result from various mechanisms such as: (1) Synthesis of transferases (acetyltransferases, phosphotransferases, nucleotidyltransferases) that modify the aminoglycoside molecule [135,136,137]. The only aminoglycoside not modified by most enzymes (except AAC(2′)-Ia,b,c) is plazomicin, but it is currently not recommended for the treatment of *S. aureus* infections [138]. (2) Lack of enzymes responsible for active transport of aminoglycosides into the bacterial cell (anaerobic metabolism of *S. aureus*, e.g., within biofilm; *S. aureus* SCV, small colony variant; *S. aureus* subsp. *anaerobius*) [139].

The most common mechanism of resistance to aminoglycosides in *S. aureus* is the synthesis of enzymes of the transferase group (Figure 1) [135,140,141,142,143,144]. The most significant are: the two-domain acetyltransferase/phosphotransferase AAC(6′)-Ie/APH(2”)-Ia, encoded by the *aacA-aphD* gene and causing resistance to gentamicin, tobramycin, kanamycin, amikacin and netilmicin, ANT(4′)-Ia nucleotidyltransferase encoded by *aadD* gene and causing resistance to tobramycin, kanamycin, neomycin and APH(3′)-IIIa phosphotransferase encoded by *aph(3*′*)-IIIa* gene causing resistance to kanamycin, neomycin, and lividomycin [141]. An APH(3′)-III phosphotransferase conditioning resistance to kanamycin and neomycin, encoded by the *aphA-3* gene (another name for the *aph(3*′*)-IIIa* gene), has also been described in vancomycin-resistant *S. aureus* (VRSA). This gene is found within the transposons Tn3851, Tn4031 and Tn5404, located on plasmids transmitted to *S. aureus* from *Enterococcus* spp. [82].

The *aacA-aphD* genes (another gene name *aac(6*′*)-Ie-aph(2”)-Ia* gene) encoding AAC(6′)-Ie acetyltransferase/APH(2”)-Ia phosphotransferase are found in Tn4001, Tn4001-like transposons located in large plasmids, e.g., pSK1 (GenBank GU565967), VRSAp present in strain Mu50 (GenBank AP003367) and in chromosomes, e.g., in SCC*mec* IV (2B&5) and, according to former nomenclature, in SCC*mec* IVc cassettes [145]. The *aacA-aphD* genes occurring together with the *ermA* methylase genes (MLS-B resistance) and the *spc* gene for spectinomicin resistance were described in the Tn6072 transposon (GenBank GU235985) [105]. AAC(6′)-Ie/APH(2”)-Ia is the only enzyme in *S. aureus* known so far to determine gentamicin resistance. The MIC of gentamicin for strains that produce this enzyme ranges from 8 mg/L to > 1024 mg/L. The MIC50 of gentamicin determined on a large population of *S. aureus* strains was 128 mg/L, and the MIC90 was 512 mg/L [141]. For strains sensitive to aminoglycosides and strains synthesizing ANT(4′)-Ia or APH(3”)-IIIa enzymes, gentamicin MIC values are in the range of 0.25–1.0 mg/L. Synthesis of AAC(6′)-Ie/APH(2”)-Ia also usually determines resistance to tobramycin. The MIC of tobramycin for strains synthesizing this enzyme, ranges from 8 mg/L to 256 mg/L (MIC50 is 32 mg/L and MIC90 is 64 mg/L) [141]. Moreover, strains synthesizing this enzyme are always resistant to kanamycin (MIC64 mg/L). The only enzyme in *S. aureus* that degrades netilmicin is AAC(6′)-Ie/APH(2”)-Ia. Netilmicin is a very weak inducer of the *aacA-aphD* gene, which is the reason that *S. aureus* strains having an inducible mechanism of regulation of this gene are often designated as sensitive in routine testing. Ida et al. [146] described natural *S. aureus* strains in which Tn4001-like elements containing the *aacA-aphD* gene retained the promoter of the beta-lactamase operon (reduced *blaZ* gene), which can result in strong induction of aminoglycoside resistance by beta-lactam antibiotics and antagonism of beta-lactams and aminoglycosides. Using aztreonam at 25 mg/L as an inducer, an increase in the MIC values of netilmicin from 4 to 32 mg/L and gentamicin from 128 to 1024 mg/L was obtained [146]. Some point mutations of the *aacA-aphD* gene can extend the spectrum of AAC(6′)-Ie/APH(2′′)-Ia by, for example, arbekacin [147].

The *aadD* gene (other gene names: *ant(4*′*)-Ia*, *aadD2*, *ant(4*′*,4*′′*)-I*) encoding ANT(4′)-Ia nucleotidyltransferase, is found in both conjugative plasmids, e.g., pGO1 of 54 kb (GenBankNC_012547) or pSK41 of 46.5 kb (GenBank AF051917), as well as in smaller non-conjugative plasmids such as pKKS825 of 14.3 kb (GenBank NC_013034) or pUB110 of 5.1 kb (GenBank AB037420). A copy of pUB110 is present in SCC*mec* type IA and type II chromosomal methicillin cassettes [29,53].

The *aph(3*′*)-IIIa* gene encoding the APH(3′)-IIIa phosphotransferase is most commonly found in plasmids and within the plasmid-embedded transposons Tn3851, Tn4031 and Tn5404 [82]. APH(3′) encoded by the *aphA-3* gene (GenBank AB300568) has also been described [117,148,149]. APH(3′)-IIIa conditions resistance to kanamycin, neomycin, paromomycin, lividomycin, livostamycin, isepamycin, butyrosin and amikacin. The high MIC of lividomycin (> 1024 mg/L) allows phenotypically distinguishing this gene from the others [146].

The *aadE* gene (other names *ant(6), ant(6)-Ia*) in plasmid pS194 encodes the ANT(6)-Ia nucleotidyltransferase that conditions streptomycin resistance [82,117].

The *aadA5* cassette gene encoding ANT(3”)-Ia nucleotidyltransferase that conditions resistance to streptomycin and spectinomycin was described in a class I integron structure (GenBank AB481128) [8,150].

The *spc* gene (another name for the *aad(9)* gene, *ant(9)-Ia*) encoding the ANT(9) nucleotidyltransferase (another name for ANT(9)-Ia) that conditions resistance to spectinomycin was described in the Tn554 and Tn6072 transposons in *S. aureus* (GenBank X02588, GU235985).

## 7. Resistance to Fluoroquinolones

Fluoroquinolones are classified as bactericidal drugs. They inhibit the activity of topoisomerase II (gyrase) and topoisomerase IV enzymes, responsible for DNA superspiralization and respiralization. According to EUCAST [16], the breakpoints for ciprofloxacin and levofloxacin are MIC > 1 mg/L, for moxifloxacin MIC > 0.25 mg/L and delafloxacin MIC > 0.25 mg/L (for skin and skin structure infections) or MIC > 0.016 mg/L (environmental pneumonia). According to CLSI [17], the limits for ciprofloxacin, levofloxacin, grepafloxacin, ofloxacin are MIC ≥ 4 mg/L and for moxifloxacin, gatifloxacin, sparfloxacin MIC ≥ 2 mg/L.

Fluoroquinolone resistance in *S. aureus* is caused by mutational changes in the *gyrA* and *gyrB* (topoisomerase II) and *parC* (*grlA*) and *are* (topoisomerase IV) genes. The mutations result in the synthesis of proteins with reduced susceptibility or insensitivity to fluoroquinolones [145]. Overproduction of the chromosome-encoded proteins responsible for efflux of fluoroquinolones from the bacterial cell (NorA, NorB, NorC and SdrM, all MFS superfamily) has also been described as a cause of the resistance [150,151]. In *S. aureus*, mutations in *gyrA* and *grlA* genes have been most frequently described. In the *gyrA* gene, the most common mutations are those causing the following substitutions in the GyrA topoisomerase II protein: S84L, A, V, or K; S85P; E86K, or G; E88V, G or K; Gl06D. In the gene encoding ParC (*grlA*), the most common mutations cause the following substitutions in the ParC topoisomerase IV protein: K23N; V4lG; R43C; I45M; A48T; S52R; D69Y; G78C; S80F, or Y; S8lP; E84K, L, V, A, G, or Y; H103Y; Al16E, or P; Pl57L; A176T, or G; N327K and P451S substitution in the ParE protein [125,152,153].

## 8. Resistance to Tetracyclines

Tetracyclines inhibit protein synthesis by interfering with the 30S subunit of the ribosome. Breakpoints of resistance to tetracycline, doxycycline and minocycline are MIC ≥ 16 mg/L according to CLSI [17] and according to EUCAST [16] for tetracycline and doxycycline MIC > 2 mg/L and minocycline and tigecycline MIC > 0.5 mg/L and MIC > 0.25 mg/L for eravacycline.

The mechanism of resistance to tetracyclines in *S. aureus* usually involves active removal of the antibiotic from the bacterial cell and ribosomal protection.

Active removal of tetracyclines from the *S. aureus* cell is conditioned by the membrane proteins Tet(K), Tet(L), Tet(38), Tet(42), Tet(43), Tet(45), Tet(63) that derive their energy from the proton pump and are classified as MFS. Most commonly, this mechanism is represented by the Tet(K) protein having 14 transmembrane segments (14 TMS) and often causes resistance to tetracyclines, except minocycline. The *tet(K)* gene has been described in plasmids pT181 (GenBank S67449) [154], pSTE2 (GenBank NC_006871), pNS1 (GenBank M16217) [155], pKH1 (GenBank U38656) [156], pKH6 (GenBank U38428) [156], pT127, pBC16. Plasmid pT181 is also found in SCC*mec* type III chromosomal cassettes [43]. Tet(L) protein having 14 TMS was described in plasmid pKKS825 (GenBank NC_014156) [116]. The gene encoding the Tet(38) protein was located in genomic DNA (GenBank AY825285) [157]. The gene encoding the 14 TSM-positive Tet(63) protein was detected in the 25664 bp plasmid pSA01-tet. The plasmid also had the genes *aacA-aphD* and *aadD* (aminoglycoside resistance) and *ermC* (MLS-B resistance) [100].

Active protection of the drug target site (ribosomal protection) involves dissociation of the tetracycline molecule from the 30S subunit of the ribosome by Tet(M) and Tet(S) proteins [82,116,158]. The Tet(M) protein determines resistance to tetracyclines including minocycline and is often the cause of resistance to tetracyclines in *S. aureus*. The *tet(M)* gene is present in the chromosome of many *S. aureus* strains, such as Mu3 and Mu50 (GenBank NC_009782, NC_002758, M21136) [120,159], and has also been described in the Tn6014 transposon (Tn5801-like) [160]. Plasmid *tet(S)* genes are characteristic of VRSA strains [82].

Resistance conditioned by Tet(U) proteins encoded by plasmids in *S. aureus*, most commonly found in VRSA strains, has also been described, but the mechanism of this resistance has not been understood to date [82]. Tetracycline resistance in *Staphylococcus* spp. conditioned by Tet(O), Tet(W) and Tet(44) proteins conditioning ribosomal protection has also been described [100].

## 9. Resistance to Mupirocin

Mupirocin inhibits protein synthesis by inactivating isoleucyl-tRNA synthetase. Topical drug used to decolonize MRSA and MSSA among healthcare personnel. No CLSI or EUCAST resistance criteria.

The cause of mupirocin resistance in *S. aureus* is the production of an isoleucyl-t-RNA synthetase encoded by the *mupA* gene (another name for the *ileS2* gene) that is insensitive to mupirocin. The *mupA* gene was described in the 37.1 kb conjugative plasmid pUSA03 (*ileS2* gene, GenBank CP000258) [112], in plasmids pJ2947 (GenBank acc. X59477, X59478) [161], MupR plasmid (GenBank DQ102365), MupR type I, II and IV plasmids (GenBank EU442885, EU442888, EU442886) [162], pJ3358, pGO400. The presence of the *mupA* gene conditions the MIC of mupirocin from 500 to >1000 mg/L [159,163,164,165,166]. A plasmid *mupB* gene of 3102 bp has also been described in *S. aureus mupB* gene shows 65.5% identity with *mupA* and 45.5% identity with *ileS* gene and conditions mupirocin resistance (MIC1024 mg/L). The plasmid containing the *mupB* gene could not be transferred to other *S. aureus* strains. The three strains in which the *mupB* gene was detected belonged to the EMRSA-2 clone (CC5/ST5) [167]. Mutations in the chromosomal *ileS* gene encoding isoleucyl-t-RNA synthetase and V58F or V631F substitutions in the IleS protein condition *S. aureus* to have reduced sensitivity to mupirocin (MIC 8–16 mg/L) [125,168].

## 10. Resistance to Fusidic Acid

Fusidic acid inhibits protein synthesis by interaction with elongation factor G (EF-G). Breakpoint of resistance to fusidic acid is MIC > 1 mg/L according to EUCAST [16]. CLSI does not provide a criterion.

Several mechanisms of resistance to fusidic acid in *S. aureus* have been described. The first mechanism is associated with mutations in the *fusA* or *fusA-SCV* genes encoding the elongation factor (EF-G). Therefore, conditioned resistance to fusidic acid is referred to as the FusA phenotype [169]. Mutations may also affect the *fusE* gene (*rplF*) encoding ribosomal protein L6 interacting with EF-G in the 30S subunit of the ribosome. The fusidic acid resistance thus conditioned is referred to as the FusE phenotype [169]. Another mechanism is related to active protection of the elongation factor. Active protection of EF-G is caused in *S. aureus* by FusB and FusC proteins encoded by *fusB* and *fusC* genes. The *fusC* gene has also been described in staphylococcal cassette chromosome (SCCfus) [170]. The FusD protein encoded by the *fusD* gene has also been described in *S. saprophiticus*. [169]. The *fusB* gene that conditions the MIC of fusidic acid from 8 to 16 mg/L was described in penicillinase plasmids in strains (FAR1 and FAR2 and FAR4 to FAR19) as early as 1974 [171], but the mechanism of resistance was understood much later. In 2002, the *fusB* gene was described in plasmid pUB101 (GenBank AY047358) [172,173] and in resistance island RASIfusB (GenBank AM292600) [174].

## 11. Resistance to Daptomycin

Daptomycin, a cyclic lipopeptide antibiotic that acts on the cytoplasmic membrane of *S. aureus*. Daptomycin aggregation with the cytoplasmic membrane is a Ca2+ ion-dependent process leading to pore formation, release of intracellular ions leading to *S. aureus* cell death. *S. aureus* strains with MIC > 1 mg/L daptomycin are defined as resistant by EUCAST [16]. Resistance or reduced sensitivity to daptomycin is probably related to mutations in the *mprF* gene encoding lysyl-phosphatidylglycerol synthetase (N352K substitution in MprF), *dltABCD* operon responsible for D-alanine attachment to teichoic acids in the cell wall (substitutions of N276S in DltB and P309L inDltD), *vraSR* regulatory genes and *clpP* (ATP-dependent Clp protease; G74S substitution), *rpoC* (RNA polymerase subunit), *vraG* (ABC transporter, permease protein), *spsB* (signal peptidase; R159S substitution), *fmtA* (autolysis and methicillin resistance-related protein), *asp23* (alkaline shock protein 23; E47K substitution in Asp23), *yycG* (synonyms: *walK, vicK*; sensor histidine kinase; substitutions: D408G, R463Q), *pgsA* (phosphatidylglycerolphosphate synthetase) [75,175,176,177,178].

## 12. Other Antibiotics

### 12.1. Resistance to Streptogramins A and Quinupristin-Dalfopristin

Streptogramins A (e.g., dalfopristin, pristinamycin) act on the 23S rRNA of the 50S subunit of the ribosome, but at a different site than MLS-B. Breakpoints of resistance are reported only for quinupristin-dalfopristin (streptogramins A/streptogramins B) and are MIC > 2 mg/L according to EUCAST [16] and MIC ≥ 4 mg/L according to CLSI [17].

Resistance to streptogramins A in *S. aureus* may result from the production of enzymes that inactivate streptogramins A such as Vat acetyltransferases encoded by the plasmid genes *vatA* [179], *vatB* located in the Tn5406 transposon [180,181], *vatC* [130], *vatD, vatE* [130] or Vgb lyases encoded by the *vgb(A*) and *vgb(B)* genes [131]. Resistance may also be conditioned by membrane proteins Vga (ATP-binding proteins) which are responsible for the efflux of streptogramin A. Vga are encoded by the plasmid and transposon genes *vga(A), vga(Av*) in Tn5406, *vga(B*), *vga(C)* and *vga(E)* in Tn6133 [180,182,183,184]. A membrane protein, Vga(A)LC, has also been described conditioning resistance to clindamycin (MIC 8–32 mg/L), lincomycin, and streptogramin A with concomitant sensitivity to erythromycin (MIC 0.094–0.19 mg/L). The *vga(A)LC* gene has been described in plasmids in *S. aureus* [124]. The production of Cfr methyltransferase that modifies the target site for streptogramin A, among others, may also be the cause of streptogramin resistance [97].

The synergistic action of streptogramins A (dalfopristin) and streptogramin B (quinupristin) causes a killing effect on *S. aureus* (including MRSA, VRSA, VISA). After quinupristin-dalfopristin treatment, a post-antibiotic effect (PAE) occurs, consisting of further inhibition of bacterial growth at drug concentrations less than the MIC. PAE values were for: *S. aureus* 2–8 h [185,186]. Resistance to quinupristin-dalfopristin in *S. aureus* is most often due to the presence of two mechanisms: constitutive resistance to MLS-B and resistance to streptogramin A. Moreover, mutation of the chromosomal gene encoding ribosomal protein L22 causes resistance of the *S. aureus* strain to quinupristin-dalfopristin [133].

### 12.2. Resistance to Rifampicin

Rifampicin inhibits transcription by interfering with the beta subunit of RNA polymerase. Breakpoint of resistance to rifampicin in *S. aureus* is MIC > 0.06 mg/L according to EUCAST [16]. CLSI reports a breakpoint of rifampin resistance in *S. aureus* (MIC ≥ 4 mg/L) [17].

Resistance to rifampicin in *S. aureus* is determined by mutations in the *rpoB* gene encoding the B subunit of RNA polymerase [187]. The most common are mutations that cause amino acid sequence changes in the RpoB protein, such as V453F, S464P, L466S, D471N A473D, A477D or T, H481N or Y, and I527L or M [125,188].

### 12.3. Resistance to Chloramphenicol

Chloramphenicol is a broad-spectrum antibacterial antibiotic that penetrates well into the cerebrospinal fluid. Due to side effects, its use has been greatly restricted. Breakpoint for resistant *S. aureus* strains is MIC ≥ 32 mg/L according to CLSI [17] or MIC > 8 mg/L according to EUCAST [16].

The cause of chloramphenicol resistance is the synthesis of chloramphenicol acetyltransferases CATA7, CATA8, CATA9 or active removal of chloramphenicol from the bacterial cell by membrane proteins belonging to the MFS superfamily CmlA1, having 12 transmembrane segments (TMS) and FexA protein with 14 transmembrane elements [189], and synthesis of adenylyl-N-methyltransferase Cfr conditioning resistance to chloramphenicol, linezolid, lincosamides, streptogramins A and pleuromutilin [97].

Resistance to chloramphenicol may also result from mutations in the *rrn5* gene, mutations in *rplC*, *rplD*, *rplV* genes encoding ribosomal proteins L3 (G152D substitution), L4 (K68Q substitution), L22. These mechanisms also determine resistance to oxazolidinones, lincosamides, streptogramins A and pleuromutilin [75]. A mutation in the *rpoB* gene (A1345G; substitution in RpoB protein D449N) was also described, conditioning the resistance of *S. aureus* strain to chloramphenicol (MIC-128 mg/L) and intermediate to oxazolidinones [94].

CATA7 acetyltransferase is encoded by *cat* genes present in plasmids pC221 (GenBank X02529) [190], pKH7 (GenBank U38429) [191], pUB112 (GenBank X02872,) [192], pSCS6 GenBank X60827) [193]. CATA8 acetyltransferase is encoded by *cat* genes present in plasmids pC223, pSCS7, pSBK203 (GenBank NC_005243, AY355285) [194,195]. CATA9 acetyltransferase encoded by *cat* genes present in plasmids pC194 (GenBank V01277), pMC524-MBM (GenBank AJ312056) [196]. The membrane protein CmlA1 (419 aa, 12 TMS) is encoded by *cmlA1* or *cmlA* genes, which have been described in *S. aureus* in class I integron (GenBank AB481130) [9]. The membrane protein FexA has been described in *S. aureus* with the ST8-MRSA-IVa/USA300 genotype (GenBank FN995110). In addition to resistance to chloramphenicol, it also conditions resistance to florphenicol [99].

### 12.4. Resistance to Fosfomycin

Fosfomycin is an inhibitor of MurA enzyme (UDP-N-acetylglucosamine-enolpyruvyltransferase) which results in inhibition of peptidoglycan synthesis. Breakpoints of resistance are MIC > 32 mg/L according to EUCAST [16].

The cause of phosphomycin resistance is the synthesis of the metalloenzyme FosB (EC 2.5.1.-) that catalyzes the Mg2+-dependent attachment of L-cysteine to the phosphomycin ring [197]. FosB has been described in the chromosomes of VISA and hVISA *S. aureus* strains Mu50 and Mu3 (GenBank NC_002758, NC_009782), among others [198].

### 12.5. Resistance to Trimethoprim

Trimethoprim is a drug used alone and in combination with sulfamethoxazole. Breakpoint of resistance is MIC > 4 mg/L according to EUCAST [16] and MIC ≥ 16 mg/L according to CLSI [17]. The cause of trimethoprim resistance is the synthesis of dihydrofolate reductase (DHFRS1) (EC 1.5.1.3) encoded in *S. aureus* by *dfrA, dfrK*, the cassette genes *dfrA12* and *dfr15* [8,9,119,120,199] and *dfrG* found in SCC*mec* [63]. The *dfrA* gene has been described in the Tn4003 transposon [197] located in plasmid pSK1 (GenBank GU565967) and in plasmids pJE1 (GenBank AF051916) and pABU1 (GenBank Y075376), among others [200]. The *dfrK* gene has been described in plasmid pKKS825 (GenBank NC_013034), pKKS627 (GenBank NC_014156) and in transposon Tn559 (GenBank FN677369) [201,202]. The cassette genes *dfrA12* and *dfr17* have been described in the class I integron structure (GenBank AB191048, AB291061) [8,9]. Substitutions of F99Y, L21Y and L41F in the DHFR protein encoded by the chromosomal *dfrB* gene may also be the cause of trimethoprim resistance [125].

## 13. Molecular Epidemiology of MRSA

The spread of MRSA is clonal. In different years, in specific geographical areas or under specific conditions (hospital, community), specific clones predominate. In general, MRSA are subdivided according to the origin of the first isolates into HA-MRSA, CA-MRSA, and LA-MRSA. Clonal affiliation, and thus presumed descent from a common ancestor, is assessed based on demonstration of similarity by several different molecular typing methods. In the case of *S. aureus* these are mainly MLST, Spa (a polymorphic VNTR in the 3′ coding region of the *S. aureus*-specific staphylococcal protein A) typing, and particularly important is the SCC*mec* cassette typing. A high correlation of sequence types with the presence of some pathogenicity genes and the type of the Agr regulatory system was observed.

The pathogenicity factors of *S. aureus* are very diverse and can be classified into several basic groups.

The first group includes pathogenicity factors that protect bacteria from the host immune system. It contains: proteins that inhibit the complement system: Sbi (*Staphylococcus aureus* binder of IgG), Efb (extracellular fibrinogen binding protein), Ecb (extracellular complement binding protein) and Scn (SCIN, staphylococcal complement inhibitor situated on a *hlb*-integrating phage SA3), chemotaxis-inhibiting protein (CHIPS) situated on a *hlb*-integrating phage SA3 [203,204], immunoglobulin binding proteins: SpA (immunoglobulin G binding protein A), Sbi (immunoglobulin binding protein), factors responsible for biofilm formation (PIA-dependent biofilm (polysaccharide intercellular antigen), PIA-independent biofilm and biofilm formed from extracellular DNA (eDNA) [205], and superantigens and superantigen-like proteins (SSL), about 40 proteins, including most enterotoxins, like-enterotoxin, toxic shock toxin TSST-1, SSL1-SSL14 proteins [206]. *S. aureus* cells are either neutralized by the defense forces of the human organism or they break these barriers and enter the phase of intensive multiplication, the so-called logarithmic growth phase (the generation time of *S. aureus* is about 20 min). After reaching an appropriate density, the bacteria stop dividing and enter the stationary phase where they start to synthesize toxins (invasive infections) or form a biofilm (chronic infections). Beta-lactam antibiotics and glycopeptides, the two most important groups of drugs in the treatment of *S. aureus* infections, have no effect on bacteria in the stationary phase. Biofilm formation by *S. aureus* hinders the penetration of drugs inside the biofilm (especially multi-molecule drugs such as glycopeptides) which results in drugs inside the biofilm not reaching the MIC let alone the MBC. Moreover, some *S. aureus* cells inside the biofilm switch to anaerobic metabolism which is associated with resistance to aminoglycosides.

The next group consists of factors enabling *S. aureus* colonization. We include here adhesive proteins such as proteins binding covalently with nasal epithelium: ClfB (clumping factor B), SasG (*S. aureus* surface protein G), Pls (plasmin sensitive protein), IsdA (iron-regulated surface determinant protein A), SdrC, SdrD (serine-aspartate repeat-containing proteins C and D); fibrinogen-binding proteins: FnbpA, FnbpB (fibronectin binding proteins A and B), ClfA, ClfB (clumping factor A and B), SdrE (serine-aspartate repeat-containing protein E), IsdA forming a covalent bond and Atl (major autolysin and adhesine) [207], Aaa (two-domain autolysin/adhesin protein) [208], Eap (extracellular adherence protein), Emp (non-covalent binding); Fibronectin-binding proteins: FnbpA, FnbpB, IsdA, (covalent binding) Atl, Aaa, Eap, Emp, Ebh (extracellular matrix-binding protein)—non-covalent binding; elastin-binding proteins: FnbpA, FnbpB (covalent binding), EbpS (elastin binding protein S)—non-covalent binding; collagen binding proteins: Cna (collagen binding adhesin)—covalent binding, Eap, Emp (non-covalent binding); bone sialoprotein covalent binding protein Bbp; glycolipid covalent binding protein Pls; vitronectin non-covalent binding proteins: Atl, Aaa, Eap, Emp; and the endoprosthesis polystyrene-binding protein Bap (biofilm-associated protein), SasC (covalent binding), Atl (non-covalent binding) [209,210]. Activation of SpA protein interaction with vWF glycoprotein may promote bacterial adhesion to damaged blood vessels [211]. This group of factors may also include products of genes encoded by ACME (arginine catabolic mobile element) that enable skin colonization.

The next group of pathogenicity factors are toxins and enzymes. Here we include hemolysins (Hla, Hlb, Hld); leukotoxins (synergohymenotropic toxins, interaction of 2 proteins one from group F and one from group S): LukF-PV/LukS-PV, HlgB/HlgA, LukF/LukS, LukD/LukE, LukF-PV/LukM, LukG/LukH; PSM (phenol soluble modulins) toxins: PSMalpha, PSMbeta; enterotoxins (SE) and enterotoxin-like (SEl): SEA-SEE, SEG-SEJ, SER-SET, SElK-SElQ, SElU-SElX; epidermolytic toxins ETA, ETB, ETD; and toxic shock toxin TSST-1 [13]. *S. aureus* also produces many enzymes and other proteins categorized as pathogenicity factors such as SplA-SplF and V8 (serine proteases and serine like proteases), ScpA (cysteine protease), Aur (zinc metalloproteinase aureolysin), Geh1 and Geh2 (Lip1 and Lip2; lipase1 and lipase2), HY-ase (hyaliauronate lyase), ceramidase [212], SrtA (sortase), and other protein factors such as Coa (coagulase, coagulation mediator), vWbp (von Willebrand binding protein) [213], EdinA-EdinC (epidermal cell differentiation inhibitor) and EDIN-like exotoxin [214]. Another group includes systems that take up iron from heme (ISD and HtsBC), from hemoglobin (ISD) and transferrin (siderophores, staphyloferrins A and B) [215,216]. Global and local regulatory systems are another group that determines the expression of many pathogenicity factors. Activity of the global Agr (accessory gene regulator, quorum sensing system)/SarA (staphylococcal accessory regulator A) system and high activity of the SaeS/R (Staphylococcus aureus exoprotein expression response regulator) system results in the synthesis of hemolysins, leukotoxins, PSM toxins (PSM1-4, PSM1-4), proteases, lipases, and proteins that protect *S. aureus* from the immune system (Sbi, Efb, Scn, Chp) and is more frequently observed in acute infections (invasive phenotype) caused by CA-MRSA.

In CA-MRSA strains SCC*mec* type IV, leukotoxin PVL and other leukotoxins are the most frequent and pathogenicity islands such as Sa1, Sa2, e.g., SR434 (ST88), Sa3, e.g., FR3757 (ST8), Sa4, e.g., MW2 (ST1) are more frequently detected, Sa, e.g., SR434 (ST88), Sa (contains leukotoxin *lukDE* genes), e.g., SR434 (ST88), Sa, e.g., SR434 (ST88), SAPISaitama (contains *tst*, *sec*, *sel* genes) (ST834) embedded in the DNA chromosome of bacteriophage SA1, SA2 (contains leukotoxin *PVL* genes) *MW2* (ST1); *FR3757* (ST8), *SA3*, the *ACME* eg FR3757 (ST8) element and many other pathogenicity genes [215,216].

Lack of Agr system activity leads to adhesion protein synthesis and biofilm formation (adhesion phenotype, chronic infections) and is characteristic of many HA-MRSA strains [8].

More important CA-MRSA clones according to [2,23,217,218,219,220,221,222,223,224,225,226,227,228,229,230,231,232] are shown in Table 4, most important HA-MRSA clones according to [23,85,125,220,224,233,234,235,236] are shown in Table 5, and most important LA-MRSA clones according to [23,125,237,238,239,240,241,242] are shown in Table 6.

## 14. Conclusions

*S. aureus* can be classified as a major human pathogen. It is also one of the microorganisms most resistant to antibacterial drugs. A particular characteristic of these bacteria is the rapid spread of clones, characterized by higher virulence and resistance to many antibiotics. One of the most important, if not the most important event in the acquisition of resistance by *S. aureus* has been the emergence of methicillin-resistant strains and the associated presence of the *mec* cassette in the genome. Although there are different variants of this element, they all greatly affect the properties of the bacteria and especially facilitate the acquisition of resistance to other antibacterial drugs. The spread of strains containing the Van operon derived from enterococci would be a particularly big problem. MRSAs containing the VanA operon have been reported in the literature, but spread has fortunately been limited to date. The possibility of acquiring enzymatic resistance to oxazolidinones (Cfr) which is determined by mobile genetic elements (plasmid, transposon) is also of great concern. The possibility of rapid emergence and spread of resistance in *S. aureus*, and often associated altered virulence, creates the need for constant monitoring of emerging bacterial variants.

## Figures and Tables

**Figure 1 ijms-23-08088-f001:**
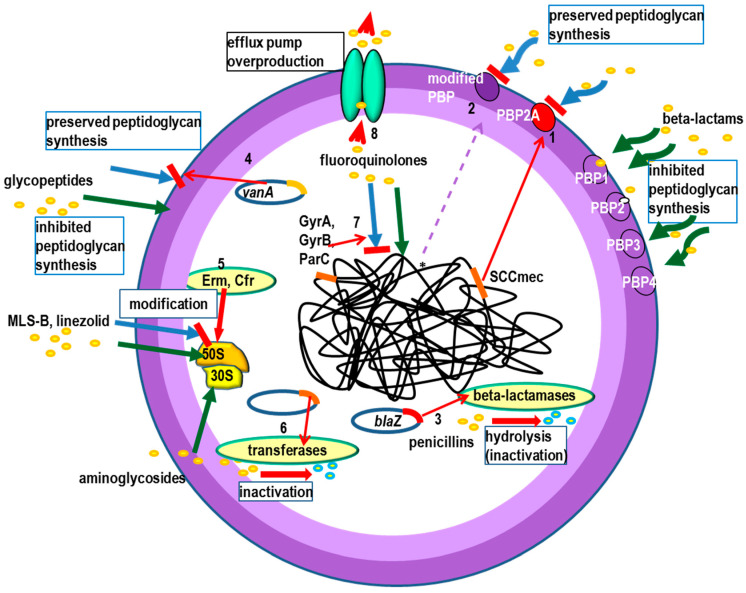
The most important resistance mechanisms in *Staphylococcus aureus*: antibiotics 
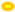
, mechanisms of action—green arrows. Resistance to beta lactams: 1. Production of penicillin-binding protein PBP2A, 2. * mutations in PBP genes—rare (MODSA), 3. beta-lactamases production -usually narrow substrate spectrum. Glycopeptide resistance: 4. VanA operon (modification of the antibiotic binding site), Linezolid resistance: 5. adenylyl-N-methyltransferase Cfr-modification 23S rRNA of bacterial ribosome. Resistance to MLS-B (macrolides, lincosamides and streptogramins B): 5. Erm—erythromycin ribosome methylation. Aminoglycosides resistance: 6. antibiotics inactivation by tansferases. Fluoroinolones resistance: 7. mutations in *gyrA* and *gyrB* (topoisomerase II) and *parC* (*grlA*) and *parE* (topoisomerase IV) genes (modification of the antibiotic binding site), 8. removal from the bacterial cell by the efflux pump.

**Figure 2 ijms-23-08088-f002:**
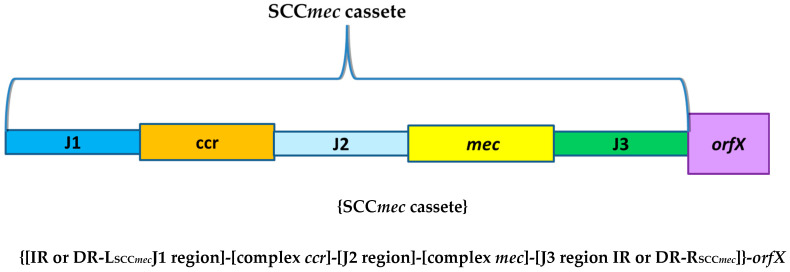
General scheme of SCC*mec* cassete.

**Figure 3 ijms-23-08088-f003:**
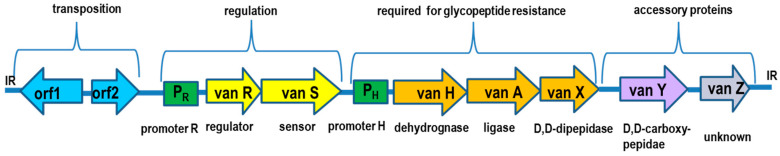
Van A operon. IR—inverted repeats, ORF—open reading frame.

**Figure 4 ijms-23-08088-f004:**
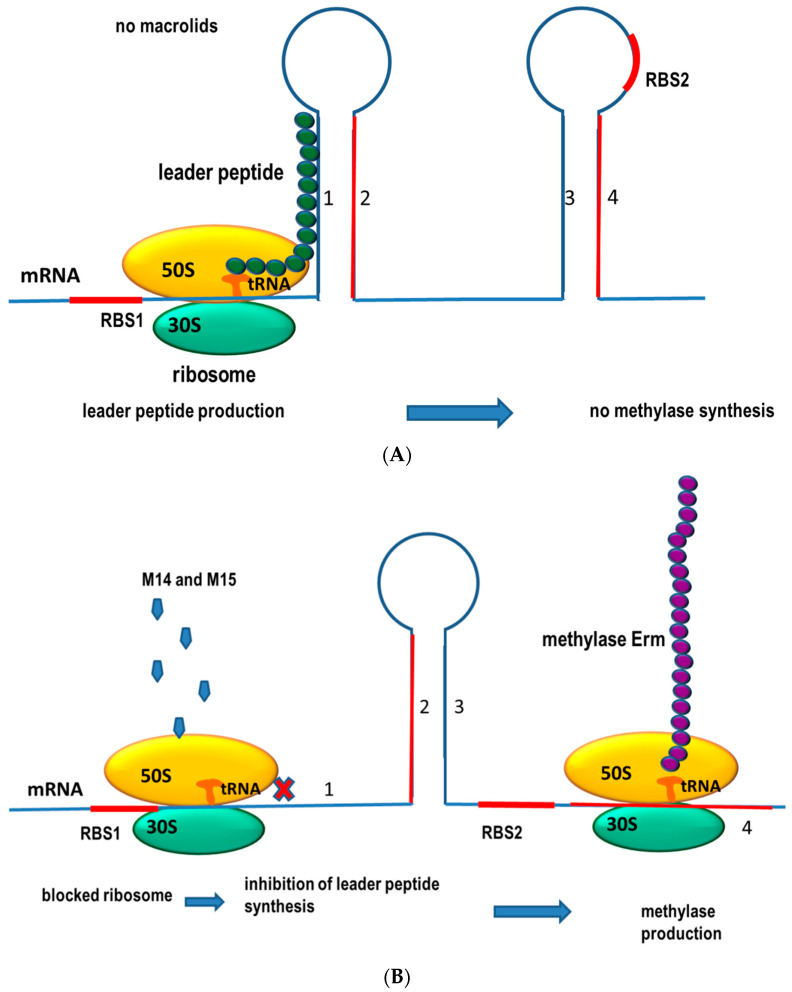
Erm methylase production by *S. aureus* is regulated at the translational level. RBS—ribosome binding site, M14 and M15—macrolides with a 14- and 15-member ring. (**A**) In the absence of M14 and M15 macrolides, a leader peptide is produced that attaches to the mRNA preventing translation of the methylase Erm (**A**,**B**). When the bacterial ribosome is blocked by macrolides, the leader peptide is not translated and the RNA conformation changes in such a way that methylase poduction is possible.

**Table 1 ijms-23-08088-t001:** The *mec* gene complexes in *S. aureus*.

Class of *mec* Gene Complex	*mec* Complex	SCC*mec* in *S. aureus*
A	IS*431*-*mecA-mecR1-mecI*	II, III, VIII, XIII, XIV
B	IS*431*-*mecA-*Δ*mecR1-*ΨIS*1272*	I, IV, VI
B2	IS*431*-*mecA*-Δ*mecR1*-Tn4001-ΨIS*1272*	IV
C1	IS*431*^→^-*mecA*-Δ*mecR1*-IS*431*^→^	VII, X
C2	IS*431*^→^-*mecA*-Δ*mecR1*-IS*431*^←^	V, IX, XII
E	*blaZ*-*mecC-mecR1_c_-mecI*_c_	XI

Abbreviations: IS-insertion sequence; IS*431*^→^ IS*431*^→^, direct repead orientation of IS; IS*431*^→^ IS*431*^←^, inverted repead orientation of IS; Tn, transposon; Δ*mecR1,* truncated *mecI-mecR; blaZ,* beta-lactamase gene; Tn4001 (transposon: *aacA-aphD,* bifunctional acetyltransferase (6′)/phosphotransferase (2″), aminoglycosides resistance determinant).

**Table 2 ijms-23-08088-t002:** *ccr* gene comlexes in *S. aureus*.

Number of *ccr* Gene Complex	Gene of *ccr*	Type of SCC*mec*
1	A1B1	I, IX
2	A2B2	II, IV
3	A3B3	III
4	A4B4	VI, VIII
5	C1 *	V, VII, XIV
7	A1B6	X
8	A1B3	XI
9	C2	XII, XIII

* 10 alleles of the C1 gene have been described. In *S. aureus*, alleles 1, 2, 3, 4, 8, 9 and 10 were described in strains JCSC3624; TSGH17; 85/2082; M; JCSC1435; P1, PM1; ZH47; M06/0171, UMCGM-4. Alleles 5 and 6, were described in *S. haemolyticus*, allele 7 in *S. epidermidis* and allele 9 in *S. saprophyticus* [39,40].

**Table 3 ijms-23-08088-t003:** Types of SCC*mec*.

SCC*mec* Type	Representative Strain	Isolated in	GenBank Accession	SCC*mec* (kb)	*ccr* Complex	*mec* Complex	Other Genes and Genetic Elements
			No.				in SCC*mec*
I	NCTC10442 (JCSC9884)	England;1961	AB033763	34.4	1	B	
II	N315 (JCSC9885)	Japan; 1981	D86934	53.0	2	A	pUB110, Tn*554*
III	85/2082 (JCSC9889)	New Zealand; 1985	AB037671	66.9	3	A	SCC_Hg_, ΨTn*554,* pT181
IV	CA05 (JCSC9890)	USA; 1999	AB063172	24.3	2	B	-
V	WIS (JCSC9897)	Australia; 1995	AB121219	27.6	5	C2	*hsdR, hsdS, hsdM*
VI	HDE288 (JCSC9900)	Portugal; 1996	AF411935	23.0	4	B	-
VII	P5747/2002 (JCSC9900)	Sweden; 2002	AB373032	32.4	5	C1	*hsdR, hsdM*
VIII	C10682 (JCSC9902)	Canada; 2003	FJ390057	32.1	4	A	Tn554
IX	JCSC6943 (JCSC9903)	Thailand; 2006	AB505628	43.7	1	C2	*arsDARBC, cadDX* *arsRBC*, *cadDX*
X	JCSC6945 (JCSC9904)	Canada; 2006	AB505630	50.8	7	C1	
XI	LGA251 (JCSC9905)	England; 2007	FR821779	29.4	8	E	*arsRBC, blaZ*
XII	BA01611	China; 2015	KR187111	49.3	9	C2	ΨSCC_BA01611_
XIII	55-99-44	Denmark; 2018	MG674089	29.2	9	A	Tn4001
XIV	SC792 (JCSC11500)	Japan; 2013–2014	LC440647	81.5	5	A	ΨSCC_pls_; ACME II’; SCC_SC640_

Abbreviations: pUB110 (plasmid: *ant(4′)*, aminoglycoside resistance; *ble*, bleomycin resistance); Tn554 (transposon: *ermA*, rRNA adenine N-6-methyltransferase, MLS-B resistance; *spc,* O-nucleotydiltransferase(9), spectinomycin resistance); ΨTn554 (transposon: *cadBC*, cadmium salt resistance); Tn4001 (transposon: *aacA-aphD*, acetyltransferase/fosfotransferase AAC(6′)-Ie/APH(2”)-Ia, aminoglycoside resistance); SCCHg (chromosomal cassette: *merRTAB*, mercury salt resistance; IS431; Tn554; *ccrC*); *hsdRSM*—endonuclease (*hsdR*), methylase (*hsdM*) genes conditioning type I modification-restriction system; *cadDX*—cadmium salt resistance genes; *arsDARBC*, *arsRBC*—arsenate resistance genes; SCCBA01611 (24. 3 kb, *ccrA1*); SCC*pls* (12 kb); ACME II’(14 kb, *arcCBDA* gene cluster, IS256); SCCSC640 (14 kb; teichoic acid bisynthesis protein F gene; *speG*, spermidine N-acetyltransferase; *ccrAB4*, chromosomal recombinase; *copA*, copper transforming ATPase).

**Table 4 ijms-23-08088-t004:** CA-MRSA clones.

CC	Clone	Spa Type	Agr Type	PVL	Other Name of Clone
1	ST1-IV/V	t125; t127; t128; t175; t273; t558; t1178; t1272; t1274 t1784; t5388	3	+/-	USA400; MW2; WA MRSA 1/45, 1/57; PFGE-1I; cMRSA; USA400 ORSA IV
1	ST772-V	t345; t345; t657; t1839; t3387; t5414; t10795;	2	+	Bengal Bay Clone; WA MRSA 60
5	ST5-IV/IV+ SCC*fus*/V/VI	t001; t002; t003; t311; t450; t1277; t2460	2	+	Peadiatric; Maltese; USA800; HDE288; Portoguese peadiatric
8	ST8-IV	t008; t024; t064; t068; t112; t121; t451; t622; t1476	1	+	USA300; USA300-0114; USA300vLA; CMRSA10; PFGE-B; CA-MRSA/J
8	ST72-IV/V	t126; t148; t324; t537; t664	1	+/-	USA700 ORSA IV;
8	ST612-IV	t1257	1	-	PFGE-A6
8	ST2021-V	t024		+	
9	ST834-IV	t1379; t9624		-	
15	ST15-IV	t084/t085	2	+	
22	ST22-IV/V	t005; t022; t032; t223; t310; t891	1	+/-	UK EMRSA-15, Barnim; PFGE-B
22	ST766-V	t1276	1	+	
30	ST30-IV	t019; t021; t318; t975; t1273	3	+	Oceania Southwest Pacific; Uruguayan 6; Mexican; USA1100; Southwest Pacific; PFGE-N; HKU-100
45	ST45-IV/V	t004/t026/t040	1		PFGE-E
59	ST59-IV/V/VII	t163; t172; t216; t316; t437; t528; t976; t3523	1	+/-	USA1000; HKU200; Western Australia MRSA-9, -15, -52, -55, -56, -73; Taiwan; Asian-Pacific, PFGE-A
59	ST87-IVb (2B)	t216	1	-	Western Australia MRSA-24
59	ST338-IV/V	t437; t441	1	+	
80	ST80-IV	t044; t131; t359; t376; t639; t1199; t1200; t1201; t1206	3	+/-	European; PFGE-G2; cMRSA
88	ST78-IV	t186; t690; t786; t1598; t2832; t3205		-	
88	ST88-IV	t168/t186/t690/t729	3	+/-	African; PFGE-J
89	ST89-IV			-	PFGE-1B
89	ST91-IV	t416/t604			PFGE-3B
93	ST93-IV	t202	3	+	Queensland; PFGE-E
121	ST121-V	t159/t314		+	
152	ST152-V	t355		+	Balkan Region
152	ST789-IV	t547		+	PFGE-1B

Abbreviations: CC, clonal compex MLST (multilocus sequence typing); ST, sequence types of MLST; Clone, sequence type of MLST -SCC*mec* type; Spa*, S. aureus* protein A; Agr, accessory gene regulator, quorum-sensing system, global regulatory system of *S. aureus*; PVL, Panton–Valentine leukocidin, +/- some variants +, and some -.

**Table 5 ijms-23-08088-t005:** HA-MRSA clones.

CC	Clone	Spa Type	Agr Type	Other Names
5	ST5-I/II	t001; t002; t003; t214; t242; t311; t586; t2460	2	UK EMRSA-3; Southern German MRSA, Rhine Hesse MRSA, Cordobes/ Chilean; PFGE-C; Geraldine; Pediatric; New York/Japan; USA100, CMRSA2; GISA
5	ST225-II	t003; t014; t151, t1282; t1623		Rhine Hesse MRSA, EMRSA-3, New York
5	ST228-I	t001; t023	2	Southern German MRSA, Rhine Hesse MRSA, EMRSA-3, New York
5	ST764-II	t002; t1064		
5	ST2590-II	t002	2	
5	ST105-II	t002	2	
8	ST8-II/IV	t008; t064; t068; t190	1	Irish-1; UK EMRSA-2/-6/; USA500 ORSA IV, USA500 ORSA II, ST8 ORSA I, ST8 ORSA IV, ST8 ORSA III; Archaic/Iberian
8	ST239-III	t030; t037	1	Hungarian; Brazilian/Hungarian; UK EMRSA-1/-4/-11; Vienna; Australian, AUS-2, AUS-3 (2000); East Australian; PFGE-B; CC8/239; ST239 ORSA III; Eurasian; Brazilian; Portuguese; PFGE-B
8	ST240-III	t037		ST240 ORSA III,
8	ST241-II/III	t037; t138	1	Finland-UK
8	ST247-I	t008; t051; t052; t054	1	Iberian, UK EMRSA-5/-7/-17; PFGE-A; ST247 ORSA I
8	ST250-I	t008; t194; t292	1/4	Archaic, ST250 ORSA I; EMRSA-8
8	ST254-I/IV	t009	1	UK EMRSA-10, Hannover MRSA
22	ST22-III/IV/V	t022; t032; t223	1/2	PFGE-B
30	ST30-I	t018; t019; t037; t268; t318	3	EMRSA-16, USA200 ORSA II
30	ST36-II	t018; t268	3	UK EMRSA-16; USA200; CMRSA4/8/9
45	ST45-II/IV	t004; t015; t026; t038; t445	1/4	USA600; CMRSA1; Berlin MRSA; USA600 ORSA II; USA600 ORSA IV
89	ST89-II	t3520	3	

Abbreviations: as in the Table 4.

**Table 6 ijms-23-08088-t006:** LA-MRSA clones.

CC	Clone	Spa Type	Agr Type	Other Name
1	ST1-IVa	t125; t127; t128; t1178	3	USA400
5	ST5-IV	t002; t003; t311	2	PFGE-I
9	ST9-III/IV/V/XII/IV+XII	t099; t100; t193; t411; t464, t526, t587, t800; t899; t1334, t1430; t2315; t2700; t3446; t4132; t4358; t4794; t13493; t29922	2	GER-MRSA-ST9, CHN-MRSA-ST9
97	ST97-IV/V	t1234	1	
97	ST1379-V	t3992	1	
130	ST130-XI	t373, t843		
398	ST398-IV/V/VII	t011; t034; t571; t1197; t1250; t1255; t1451; t1456; t1928; t2510	1	GER-MRSA-ST398, CHN-MRSA-ST398
398	ST1232-V	t034		

Abbreviations: as in the Table 4.

## Data Availability

Not applicable.

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
