# Peer review of "Molecular Mechanisms of Drug Resistance in Staphylococcus aureus"

_ijms, 2022, doi:10.3390/ijms23158088_

Round 1
Reviewer 1 Report
This is review of molecular mechanisms of drug resistance in Staphylococcus aureus by Mlynarczyk-Bonikowska. Drug resistance in S. aureus is an important public issue. The review details molecular mechanisms of more than ten (classes of) antibiotics resistance in S. aureus. It is an interesting and thorough review. The manuscript is concisely and clearly written. The paper will offer useful information to the readers. The manuscript will be suitable for publication, and as such my comments are limited.
There are a few minor points, however, that require the authors’ attention.
11. Page 2. Line 69-74, “Resistance to glycopeptides encoded by the VanA operon (usually vancomycin MIC≥16 mg/L was expressed only in S. aureus strains with a mutant modification-restriction system… Therefore, only a dozen VRSA strains with the VanA operon have been described worked wide (4,5).” This statement is not correct or at least it is overstated. The data from reference 4 and 5 did not concluded or supported above statement. Please reword the sentence.
22. Page 9. Line 325, “It was found that all VRSA strains belonged to a common MLST CC5 clonal complex.” This is not right. The fact is: the majority of VRSA strains (12 out 13) isolated in the U.S. belonged to clonal complex 5. Please see following references:
Limbago BM, et al., Journal of Clinical Microbiology. 2014. 52:998-102.
McGuinness WA, et al., Yale Journal of biology and Medicine. 2017.90:269-181.
33. Page 9. Lane 326-328. “In addition, all VRSA with the VanA operon had a mutation in the hsdR gene encoding Sau1 (1 modification-restriction system). In strains lacking this mutation, the vanA operon was not expressed. (5).” Again, this statement is not right. Please revise the sentence. Reference 5 has no data to support the above statement or conclusion. Reference 5 only mentioned hsdR gene (Sau1 system) in the discussion. Whether the VRSA strains have similar mutations in the Sau1 type 1 restriction-modification system has been investigated.
Author Response
Thank you very much for your valuable comments. Referring to them:
- Page 2. Line 69-74, “Resistance to glycopeptides encoded by the VanA operon (usually vancomycin MIC≥16 mg/L was expressed only in S. aureus strains with a mutant modification-restriction system… Therefore, only a dozen VRSA strains with the VanA operon have been described worked wide (4,5).” This statement is not correct or at least it is overstated. The data from reference 4 and 5 did not concluded or supported above statement. Please reword the sentence.
Is changed to:
"Resistance to glycopeptides encoded by the VanA operon (usually vancomycin MIC≥16 mg/L) was expressed more frequently in S. aureus strains with mutation of the modification-restriction system and/or possessing the pSK41-like conjugation plasmid (factors that increase the frequency of VanA operon conjugation). Therefore, only about a dozen VRSA strains with the VanA operon have been described on the working scale."
- Page 9. Line 325, “It was found that all VRSA strains belonged to a common MLST CC5 clonal complex.” This is not right. The fact is: the majority of VRSA strains (12 out 13) isolated in the U.S. belonged to clonal complex 5. Please see following references:
Changed to:
“Most VRSA strains have been found to belong to the common MLST CC5 clonal complex”
We have also included in the literature:
- Limbago, B.M.; Kallen, A.J.; Zhu, W.; Eggers, P.; McDougal, L.K.; Albrecht, V.S. Report of the 13th vancomycin-resistant Staphylococcus aureus isolate from the United States. Clin. Microbiol. 2014, 52, 998-1002; doi: 10.1128/JCM.02187-13
- McGuinness, W.A.; Malachowa, N.; DeLeo F.R. Vancomycin resistance in Staphylococcus aureus
. Yale J. Biol. Med. 2017,90, 269-281
- Page 9. Lane 326-328. “In addition, all VRSA with the VanA operon had a mutation in the hsdR gene encoding Sau1 (1 modification-restriction system). In strains lacking this mutation, the vanA operon was not expressed. (5).” Again, this statement is not right. Please revise the sentence. Reference 5 has no data to support the above statement or conclusion. Reference 5 only mentioned hsdR gene (Sau1 system) in the discussion. Whether the VRSA strains have similar mutations in the Sau1 type 1 restriction-modification system has been investigated.
Changed to:
"In addition, some VRSA with the VanA operon had a mutation in the hsdR gene encoding Sau1 (1 modification-restriction system). "The next sentence is removed.
Reviewer 2 Report
the paper describe the molecular mechanisms of drugs resistance in S. aureus. The paper is well written and esaustive, but need some concerns before publication.
- the name of the gens and the name of the species should be written in italics all over the text.
-lines 225-225 it should be replace "was" with "were".
lines 243-254: the informations reported in this sentence were the same of table 3, it should be better eliminate this part or report as part of the legend.
- lines 296 " ' " should be not use.
-line 309 there is a "." that should be eliminated and the sentence revised.
- line 367 "a MIC" not "an MIC"
- line 429 Clostridium not "C.", is the first time that appeared in the text.
- lines 608-641 why the different alleles of tet genes were reported in (...)?
-lines 724 and 729 "B" should be replaced with "beta" or the symbol of "beta"
- lines 787-789 the authors could be eliminate the explanation of HA-MRSA, CAMRSA and LA-MRSA.
-line 791 this sentence needs a full stop after "molecular typing"
- line 791-794 MLST should be reported in (...) and after Spa and SCCmec cassette should be add "typing".
-lines 789-799 this sentence should be revised the verb is at plural.
- lines 873-876 "more" should be replaced with "most" .
- figure 1 the image should be ameliorated especially in the part wrote in boxes. The explanation lines 111-119 should be wrote in the legend and revised, this part is not clear to the reader. line 113 correct "resistsnce".
Author Response
Thank you very much for your valuable comments. Referring to them:
Thank you very much for your valuable comments
1. the name of the gens and the name of the species should be written in italics all over the text.
We corrected it
-2 lines 225-225 it should be replace "was" with "were".
Replaced
3. lines 243-254: the informations reported in this sentence were the same of table 3, it should be better eliminate this part or report as part of the legend.
In the text we list the subtypes while the table contains the types. Optionally, we can move this piece of text to the legend.
-4.lines 296 " ' " should be not use.
Removed
5. line 309 there is a "." that should be eliminated and the sentence revised.
Changed to:
“The VanA operon that determines resistance to vancomycin (MIC 64-1024 mg/L) and teicoplanin (MIC 16-512 mg/L) is composed of 7 genes, (vanRASAHAAXAYAZA) (Fig3) located at Tn1546 and was described in E. faecalis, E. faecium, E. gallinarum, E. casseliflavus, E. avium, E. durans, E. mundtii and E. rafinosus [76].). The Van A operon shows inducible expression mediated by two regulatory genes vanRA (regulator) and vanSA (sensor, a signal histidine kinase located in the cytoplasmic membrane). vVanSA sensor activation is caused by both vancomycin and teicoplanin.”
6. line 367 "a MIC" not "an MIC"
Corrected
-7. line 429 Clostridium not "C.", is the first time that appeared in the text.
Corrected
-8 lines 608-641 why the different alleles of tet genes were reported in (...)?
The way the tet genes were transcribed we repeated after other authors. It seems that for specifically the tet genes in S. aureus, such a transcript is most commonly used
-9 . lines 724 and 729 "B" should be replaced with "beta" or the symbol of "beta"
Replaced
-10. lines 787-789 the authors could be eliminate the explanation of HA-MRSA, CAMRSA and LA-MRSA.
Eliminated
-11. line 791 this sentence needs a full stop after "molecular typing"
Corrected
- 12. line 791-794 MLST should be reported in (...) and after Spa and SCCmec cassette should be add "typing".
Added
-13.lines 789-799 this sentence should be revised the verb is at plural.
Corrected ("The first group includes pathogenicity factors that protect bacteria from the host immune system. It contains:.....)
14.- lines 873-876 "more" should be replaced with "most" .
Replaced
-15. figure 1 the image should be ameliorated especially in the part wrote in boxes. The explanation lines 111-119 should be wrote in the legend and revised, this part is not clear to the reader. line 113 correct "resistsnce".
The figure and the text is corrected